# Insomnia as a Behavioral Pathway from Fear of Missing Out to Depression in Emerging Adults

**DOI:** 10.3390/brainsci15090917

**Published:** 2025-08-26

**Authors:** Brian N. Chin, Yuxi Xie

**Affiliations:** 1Department of Psychology, Trinity College, Hartford, CT 06106, USA; 2Department of Psychology, University of Macau, Taipa, Macau 999078, China; yuxixie@um.edu.mo

**Keywords:** emerging adulthood, gender differences, mental health, psychological mechanisms, sleep, young adulthood

## Abstract

Background/Objectives: Fear of missing out (FOMO) refers to the pervasive experience of worrying that others may be having rewarding or meaningful experiences from which one is absent or excluded. FOMO has been linked with both sleep disturbances and poor mental health outcomes, particularly in emerging adults (ages 18–29). This study tested whether insomnia symptoms mediate the relationship between FOMO and depressive symptoms in emerging adults and whether gender moderates the links between FOMO, insomnia symptoms, and depression symptoms. Methods: We conducted a secondary analysis of cross-sectional survey data from 849 emerging adults in the United States. Participants completed validated measures of FOMO, insomnia symptoms, and depression symptoms. We tested our hypotheses using regression models in SPSS version 29 and mediation and moderation models via the PROCESS macro. Analyses included age, race/ethnicity, and education as covariates. Results: FOMO predicted greater insomnia severity and more depression symptoms, and insomnia severity partially mediated the link between FOMO and depression symptoms. The FOMO–insomnia association was moderated by gender, with a stronger link among men. Conclusions: These findings suggest that insomnia is a plausible mechanism linking FOMO to depression in emerging adults. Gender differences suggest that FOMO may disproportionately disrupt sleep in men and highlight the need for tailored prevention efforts to target both FOMO and sleep disruption among emerging adults.

## 1. Introduction

Fear of missing out (FOMO) refers to the pervasive experience of worrying that others may be having rewarding or meaningful experiences from which one is absent or excluded [1]. FOMO can be understood as a motivational–affective state characterized by heightened sensitivity to potential social exclusion and reward deprivation. It involves a persistent attentional focus on others’ activities, combined with emotional distress and compensatory behaviors aimed at social re-engagement. FOMO is a common and developmentally salient experience, particularly during the transition to adulthood. Approximately half of U.S. adults experience FOMO at least monthly, with emerging adults reporting more frequent and intense FOMO [2]. A growing body of research has begun to identify the affective and behavioral consequences of this experience. There is growing evidence linking FOMO to a range of negative outcomes, including psychological distress and physical health complaints [3], lower perceived social support and more unmet psychological needs [4], and poorer sleep quality [5].

Emerging adulthood spans the ages of 18 to 29 and is a critical developmental period marked by identity exploration, heightened social comparison, and uncertainty about life direction—all of which may intensify vulnerability to FOMO-related distress [6]. In this context, FOMO may represent not only a reaction to social exclusion but also a signal of chronically unmet relational and belongingness needs, particularly in peer and group contexts. The emotional salience of FOMO during this life stage may be further exacerbated by other age-related vulnerabilities, such as elevated risk for disrupted sleep [7] and changing social networks [8], each of which is independently associated with poorer mental health outcomes [9,10]. Although FOMO is increasingly recognized as a digital-age stressor, most studies have examined its associations with problematic social media use [11], rather than testing the behavioral mechanisms through which FOMO influences mental health, such as insomnia, which is causally linked with depression risk [9,12]. To our knowledge, no studies have directly tested insomnia as a behavioral mechanism linking FOMO to depression. Moreover, little is known about whether these effects differ by individual factors such as gender. This study aimed to address these gaps by testing whether insomnia symptoms mediate the relationship between FOMO and depression symptoms in a sample of U.S. emerging adults and whether gender moderates the strength of these associations.

Several complementary theoretical frameworks help explain why FOMO arises and how it functions as a socially driven motivational–affective state with behavioral and psychological consequences. We use these frameworks as conceptual scaffolds for a testable mechanistic pathway in which FOMO is associated with greater insomnia, which in turn relates to higher depressive symptoms; we also examine whether this pathway varies by gender. From the perspective of attachment theory, humans have a biobehavioral drive to seek attachment security through interactions with close others [13]. When this need is thwarted, such as in the case of perceived social exclusion, individuals may experience FOMO as a signal that their social bonds are insecure. This sense of insecurity may heighten vigilance toward others’ activities, leading to preoccupation, emotional distress, and behaviors aimed at re-establishing inclusion. Critically, this hypervigilant state may also interfere with one’s ability to relax and disengage from social monitoring at bedtime, contributing to pre-sleep arousal and insomnia symptoms. In turn, insomnia is a well-documented risk factor for depression, particularly during emerging adulthood, a developmental period when relational bonds and social status are particularly salient for identity formation and emotional regulation [9,12].

Social Baseline Theory offers a neurobiological perspective that complements these views by proposing that humans are evolutionarily adapted to expect the presence of supportive social relationships as a baseline condition for a safe environment [14]. When this baseline expectation is violated, such as during the experience of FOMO, greater regulatory effort is required to manage threat, uncertainty, and emotional distress. From this perspective, FOMO may not only reflect perceived social threat but also signal a violation of the brain’s default assumption of social proximity and support. This violation may increase activation in threat-related neural circuits and heighten autonomic arousal, including elevated heart rate or vigilance, which can interfere with the downregulation needed to initiate and maintain sleep. Over time, these disruptions to regulatory systems and sleep processes may contribute to a greater risk of depression.

Self-determination theory provides a complementary psychological lens that posits that well-being depends on satisfying the three core needs of autonomy, competence, and relatedness [15]. In theory, FOMO may emerge when relatedness needs are unmet, triggering a loop of psychological and physiological arousal (e.g., anxiety, vigilance) and behavioral compensation (e.g., excessive social monitoring) aimed at re-establishing social inclusion. This hyperarousal and social monitoring may extend into nighttime hours, interfering with sleep onset and maintenance. Indeed, FOMO has been associated with difficulty disengaging from digital or social stimuli before bed, leading to elevated pre-sleep cognitive arousal and disrupted sleep [2]. Over time, chronic FOMO may undermine mental health by reinforcing anxious preoccupation with others and disrupting sleep–wake rhythms. This is particularly salient during emerging adulthood, when daily routines and social schedules are often irregular or still being established [16]. Sociometer theory similarly conceptualizes FOMO as an indicator of social threat—when individuals perceive that they are being excluded by others, this perceived threat to belonging may lower self-esteem, increase psychological distress, and lead to evolutionarily conserved behavioral responses aimed at repairing social status or inclusion [17]. Thus, both self-determination theory and sociometer theory regard FOMO as a motivational signal that initiates behavioral and emotional regulation—some of which may involve sleep disruption that can subsequently increase vulnerability to depression.

These frameworks collectively suggest that FOMO is not merely a byproduct of digital technology use but a relationally and developmentally embedded psychological phenomenon that is anchored in the human need for connection and belonging and shaped by evolutionary pressures toward social monitoring and inclusion. Moreover, they suggest that FOMO may have motivational and behavioral consequences that impact health and well-being, with some accounts further proposing that these impacts may differ by gender. For instance, the tend-and-befriend model posits that women are more likely to cope with stress by seeking affiliation and support [18]. According to this model, it is also plausible that women may respond to FOMO by prioritizing reconnection, whereas men may respond by withdrawing or emotionally disengaging. These diverging strategies for managing social threat may lead to FOMO having a different impact on the sleep and mental health of men versus women. However, we are unaware of earlier work testing whether the connections between FOMO, sleep, and mental health are moderated by gender.

The present study tested whether insomnia symptoms mediate the relationship between FOMO and depression symptoms among emerging adults in the United States (Aims 1–2). We also examined whether gender moderates these associations (Aim 3). We hypothesized that FOMO would be associated with greater insomnia severity and more depression symptoms and that insomnia symptoms would mediate the relationship between FOMO and depression symptoms. In addition, we explored whether the strength of these associations varied by gender.

## 2. Materials and Methods

This was a secondary analysis of data collected for a cross-sectional study of social media and sleep [19]. A sample of emerging adults was recruited via Prime Panels [20]. Eligibility criteria were age 18–30, residence in the United States, and English fluency; there were no additional inclusion or exclusion criteria. Although emerging adulthood is typically defined as spanning ages 18–29, we retained 30-year-old participants because the parent study recruited participants aged 18–30. Procedures were approved by Trinity College’s Institutional Review Board (ID: 3172). Data were collected between 20 and 22 February 2024.

After providing informed consent, participants completed a Qualtrics survey assessing their FOMO experiences, insomnia symptoms, depression symptoms, and demographic characteristics. Participants were compensated upon completing this study.

### 2.1. Measures

The 10-item Fear of Missing Out Scale [1] assessed the extent to which participants generally experienced apprehension about being left out of social events and rewarding experiences (e.g., “I fear others have more rewarding experiences than me”). Items were rated on five-point scale and averaged to create a total FOMO score; higher scores indicate greater FOMO.

The 7-item Insomnia Severity Index [21] assessed whether participants had difficulty falling or staying asleep during the past two weeks. Items were rated on a five-point scale and averaged to create a total insomnia symptom severity score; higher scores indicate greater insomnia severity.

The 10-item Center for Epidemiological Studies Depression Scale [22] assessed participants’ feelings and behaviors during the past week. Items were rated on a four-point scale and summed to create a total depression symptom score; higher scores indicate greater depressive symptoms.

Finally, participants completed a demographic questionnaire assessing their age, gender, race/ethnicity, and educational attainment.

### 2.2. Data Analysis

Prior to hypothesis testing, we calculated the internal consistency of the FOMO, insomnia severity, and depression symptom scales. Next, we tested each aim using the PROCESS macro for SPSS (Version 4.2) [23] with age, race, and educational attainment included as covariates in all models. Aim 1 and Aim 2 were tested using Model 4 (simple mediation) to estimate the indirect effect of FOMO on depressive symptoms via insomnia symptoms using bias-corrected bootstrapping with 10,000 resamples. Indirect effects were considered statistically significant when the 95% bias-corrected bootstrapped confidence interval did not include zero. Aim 3 was tested using Model 1 (moderation) to examine whether the associations of FOMO with insomnia symptoms and with depressive symptoms differed by gender. We probed the simple slopes and visualized the interaction. Post hoc sensitivity analyses used Model 7 (moderated-mediation) to test whether gender moderated the a-path (FOMO–insomnia) of the indirect effect. We report conditional indirect effects of FOMO on depressive symptoms via insomnia severity for females and males, as well as the index of moderated mediation.

## 3. Results

Our sample consisted of 849 participants between the ages of 18 and 30 (M = 24.3, SD = 3.9). The majority identified as female (63.0%), followed by male (35.8%) and non–binary (1.2%). The sample’s racial and ethnic identification was 53.5% White, 28.6% Black or African American, 15.7% Hispanic or Latino, 4.9% Asian or Asian American, 2.9% American Indian or Alaska Native, 0.4% Native Hawaiian or Other Pacific Islander, and 0.7% another race/ethnicity. Participants reported a range of educational attainment levels: 6.8% had less than a high school diploma, 41.5% had a high school diploma or equivalent, 24.5% had completed some college without earning a degree, 9.8% had an associate’s degree, 14.5% had a bachelor’s degree, and 2.9% had a graduate degree.

All multi-item scales demonstrated good internal consistency with Cronbach’s α values of 0.89 for FOMO, 0.86 for insomnia symptoms, and 0.82 for depression symptoms. On average, participants reported moderate levels of FOMO (M = 2.6, SD = 0.9, range = 1–5), insomnia symptoms (M = 11.4, SD = 6.1, range = 0–28), and depression symptoms (M = 13.2, SD = 6.1, range = 0–30). Zero-order correlations among the key study variables were all positive and statistically significant: FOMO was correlated with insomnia symptoms (r = 0.46, *p* < 0.001) and depression symptoms (r = 0.49, *p* < 0.001), and insomnia symptoms were correlated with depression symptoms (r = 0.60, *p* < 0.001). Table 1 shows demographic characteristics and means and standard deviations for key study variables stratified by gender.

### 3.1. FOMO, Insomnia, and Depression: Mediation Analysis (Aims 1–2)

We used Model 4 of the PROCESS macro to test whether FOMO was associated with insomnia and depression symptoms and whether insomnia symptoms mediated the relationship between FOMO and depression symptoms.

Consistent with our first hypothesis, FOMO was associated with greater insomnia severity (B = 3.15, SE = 0.21, t = 14.97, *p* < 0.001, 95% CI [2.73, 3.56], β = 0.46) and also predicted higher depression symptoms (B = 3.34, SE = 0.20, t = 16.39, *p* < 0.001, 95% CI [2.94, 3.74], β = 0.49).

Analyses of total, direct, and indirect effects are summarized in Figure 1. Consistent with our second hypothesis, the indirect effect of FOMO on depression symptoms via insomnia severity was significant (B = 1.49, SE = 0.13, 95% CI [1.23, 1.77], β = 0.22, BootSE = 0.02, 95% CI [0.19, 0.26]). The direct effect of FOMO on depression remained significant after controlling for insomnia severity (B = 1.85, SE = 0.20, 95% CI [1.46, 2.24]).

### 3.2. Moderation by Gender (Aim 3)

Due to the small number of participants identifying as non–binary (*n* = 10), gender moderation analyses were limited to participants who identified as male (*n* = 304) or female (*n* = 535). We used Model 1 of the PROCESS macro to test whether gender moderated the relationship between FOMO and (a) insomnia severity and (b) depression symptoms.

As shown in Figure 2, there was a significant interaction between FOMO and gender predicting insomnia severity (B = 1.50, SE = 0.43, *p* < 0.001). Analysis of conditional effects indicated that the association of FOMO and insomnia severity was stronger among males (B = 4.08, SE = 0.33, t = 12.32, *p* < 0.001, 95% CI [3.43, 4.73]) than females (B = 2.58, SE = 0.27, t = 9.52, *p* < 0.001, 95% CI [2.05, 3.11]). However, the interaction between FOMO and gender was not significant when predicting depression symptoms (B = 0.54, SE = 0.41, *p* = 0.19), indicating that this association was not moderated by gender.

### 3.3. Moderated Mediation (Sensitivity Analysis)

In exploratory, post hoc moderated mediation analyses, we used Model 7 of the PROCESS macro to test whether the indirect effect of FOMO on depressive symptoms via insomnia symptoms varied by gender. In this model, gender moderates only the a-path (FOMO–insomnia severity).

The conditional indirect effect was significant for females (ab = 1.22, 95% CI [0.92, 1.54]) and for male (ab = 1.93, 95% CI [1.57, 2.31]), with a larger indirect effect for males; the index of moderated mediation was 0.71 (95% CI [0.31, 1.12]). These findings converge with the primary mediation and moderation results.

## 4. Discussion

This study investigated whether, how, and for whom FOMO is associated with disturbed sleep and mental health in American emerging adults. As predicted, FOMO was associated with greater insomnia severity and higher depression symptoms. The standardized effect sizes for these associations were in the medium-to-large range, indicating that these impacts are notable and may have clinical (public health) relevance. Moreover, the association between FOMO and depression symptoms was partially mediated by insomnia severity, and the link between FOMO and insomnia severity was stronger among male emerging adults than female emerging adults; the relationship between FOMO and depression symptoms was not moderated by gender.

Across our guiding frameworks of attachment theory, self-determination theory, and sociometer theory, FOMO can be understood as a psychological signal that one’s relatedness or belongingness needs are threatened or unfulfilled. Each of these frameworks converges on the idea that humans are biologically wired to monitor for and respond to cues of social exclusion, with the goal of restoring belongingness and social connection. Social Baseline Theory further complements this view by proposing that the brain assumes the availability of social support as the default condition for minimizing cognitive, metabolic, and emotional demands [14]. In this view, FOMO is more than just momentary anxiety about being left out—it reflects a motivational and affective state that elicits compensatory behaviors and psychological distress when social needs are unmet.

The observed associations between FOMO, insomnia, and depression symptoms are consistent with this theoretical framing. Heightened sensitivity to social exclusion triggered by FOMO may also activate physiological arousal that impairs sleep. From a neurobiological perspective, the violated expectation of social proximity may increase self-regulatory load, promoting vigilance and disrupting the physiological downregulation necessary for initiating and maintaining sleep [24]. In turn, both persistent social disconnection and impaired sleep may detrimentally impact mental health and emotion regulation over time. The observed partial mediation by insomnia suggests that sleep disruption may represent one pathway through which FOMO has downstream influences on mental health. This is especially plausible in an emerging adulthood context, given that this life stage is characterized by changing routines and social networks [8,16] and heightened sensitivity to social exclusion [25]. Other candidate pathways that should be explored in future work include FOMO’s impact on social behaviors and cognitions [26] and emotion regulation processes [27].

From an evolutionary perspective, these findings are consistent with the idea that FOMO may function as a social monitoring adaptation. Because humans evolved in interdependent groups and exclusion posed a genuine survival risk, the distress of FOMO may serve a protective function by alerting individuals to potential disconnection and motivating reconnection efforts. However, this signal may become chronically activated in a modern social context where around-the-clock connectivity facilitated by digital technology creates a constant opportunity for social comparison and perceived exclusion. In the absence of protective social cues, the brain may interpret persistent signals of exclusion, such as FOMO, as threats to safety, leading to prolonged physiological arousal, disrupted sleep, and psychological dysregulation. Recent experimental evidence suggests that individuals with insomnia may exhibit heightened vulnerability to depressed mood following immune system activation, even in the absence of elevated inflammation levels [28]. Thus, it is possible that FOMO-related sleep disruptions could sensitize emerging adults to inflammatory triggers that contribute to depression symptoms.

Finally, the gender moderation observed for insomnia suggests that individual differences in coping responses to perceived social threat may shape the extent to which FOMO impacts health and well-being. FOMO was associated with greater insomnia severity for both males and females, with a stronger association among males. Although speculative, these findings are consistent with the “tend-and-befriend” model, which proposes that women are more likely to manage social stressors through affiliative behaviors, such as support-seeking [18]; men, in contrast, are more likely on average to respond to perceived exclusion with withdrawal or disengagement, potentially leaving arousal less buffered at bedtime and increasing their risk for sleep disruption. Future research should test whether the observed moderation effect can be explained by gender differences in emotion regulation [29], social behavior, or qualitative differences in how FOMO is experienced, appraised, or caused in men and women. In contrast, we did not observe gender moderation for the link between FOMO and depressive symptoms. One possibility is that gender differences are more relevant proximally during pre-sleep arousal, whereas depressive symptoms reflect multiple additional influences (e.g., broader stress exposure, emotion regulation, social support) that attenuate the moderating role of gender. Future work should test this explanation, for example, by comparing male and female emerging adults’ accounts of how FOMO is appraised and managed and by evaluating multi-mediator and serial mediation models.

These findings align with neurobiological models that emphasize the importance of social regulation in reducing cognitive and emotional effort [14]. Within this framework, restoring a sense of social safety may reduce the physiological costs of FOMO and improve sleep. If FOMO reflects a motivational signal of unmet social needs, interventions that can reduce excessive social vigilance or augment social belonging, such as mindfulness training [30], may help buffer its impact on sleep and depression. Given that insomnia partially mediated the link between FOMO and depressive symptoms, we view sleep as a modifiable pathway through which FOMO may confer risk for depression. Thus, sleep-focused interventions (e.g., cognitive behavioral therapy for insomnia, circadian stabilization) would be expected to reduce the downstream depression symptoms associated with FOMO. This implication is consistent with evidence that insomnia treatment can reduce depression risk [31]. Such approaches may be especially beneficial during developmental risk periods, such as emerging adulthood, when FOMO is especially salient and both sleep patterns and social routines are still being established.

Strengths of this investigation include its demographically diverse sample of U.S. emerging adults and use of validated assessments of FOMO, insomnia, and depression symptoms. Moreover, the inclusion of both mediation and moderation analyses to examine underlying processes and individual differences extends previous correlational evidence by clarifying how and for whom FOMO may be most disruptive.

Several limitations also warrant consideration and future research to address them. First, the cross-sectional design of this investigation prevents us from drawing strong causal inferences about the associations we observed in this study. Although findings were consistent with our hypothesized process-based model and earlier experimental evidence suggests that sleep disturbances are causally linked with depression risk [9,12], the directionality of the links between FOMO, sleep disturbance, and depression in this emerging adulthood context remains unclear. Future work could address this limitation by using longitudinal or experimental designs to clarify the temporal relationship of the associations observed in this study, such as by testing whether interventions aimed at reducing FOMO in emerging adults can improve sleep and mental health.

Second, our self-report assessments of FOMO, insomnia symptoms, and depression symptoms introduce the possibility of shared method variance, such as common mood states or dispositional factors influencing all measure responses. Nonetheless, prior research suggests that the associations between social experiences and sleep disruption often persist even when controlling for neuroticism [32]. Future research could address this limitation by incorporating a clinician-rated diagnostic assessment of insomnia and examining objective, ecologically valid sleep measures (e.g., actigraphy, polysomnography, daily sleep diaries) as complementary indices of sleep timing and continuity.

Third, our sample included a relatively small number of non-binary participants, which limited our ability to test for gender moderation beyond a binary framework. Given the likely underrepresentation of non-binary individuals in our sample [33], future studies should prioritize the recruitment of more gender-inclusive samples to examine the impacts of FOMO for non-binary emerging adults. It is possible that FOMO may be especially salient in this population due to heightened threats of exclusion and identity-based marginalization. Future studies could explore whether unique social stressors or coping strategies among non-binary individuals modify the pathways between FOMO, sleep, and depression. Moreover, we did not assess certain demographic and health indicators (e.g., marital status, occupation, diagnosed health conditions, medication use), which limited our ability to evaluate these potential confounders.

Fourth, this study focused on insomnia as a behavioral mechanism linking FOMO to depression, but other processes may also contribute to this association, especially since we observed partial mediation with a residual direct effect. For example, emotion regulation difficulties, patterns of social media engagement, and perceived social support may also mediate the FOMO–insomnia relationship. At a neurobiological level, future studies should test whether FOMO is associated with altered functioning in brain systems involved in social threat detection, reward sensitivity, or cognitive control. For instance, prior research suggests that exclusion-related distress activates the anterior cingulate cortex and insula, regions implicated in social pain and sleep disruption [34,35]. Future research should more comprehensively examine FOMO’s biopsychological mechanisms to inform more tailored intervention efforts.

## 5. Conclusions

This study implicates insomnia as a potential behavioral pathway linking FOMO to depression symptoms in emerging adults and suggests possible gender differences in vulnerability to the sleep disturbances associated with FOMO. Findings highlight the impact of FOMO on sleep and mental health and suggest that behavioral health efforts aimed at improving sleep and reducing FOMO may be especially impactful during this developmental stage, especially among male emerging adults.

## Figures and Tables

**Figure 1 brainsci-15-00917-f001:**
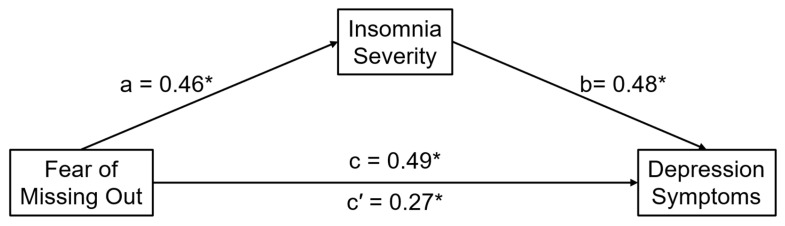
Conceptual mediation model testing the indirect effect of FOMO on depression symptoms via insomnia symptom severity. Path labels reflect the age-adjusted total (c), direct (c′), and indirect (a × b) effects. All coefficients are standardized. Asterisk indicates *p* < 0.05.

**Figure 2 brainsci-15-00917-f002:**
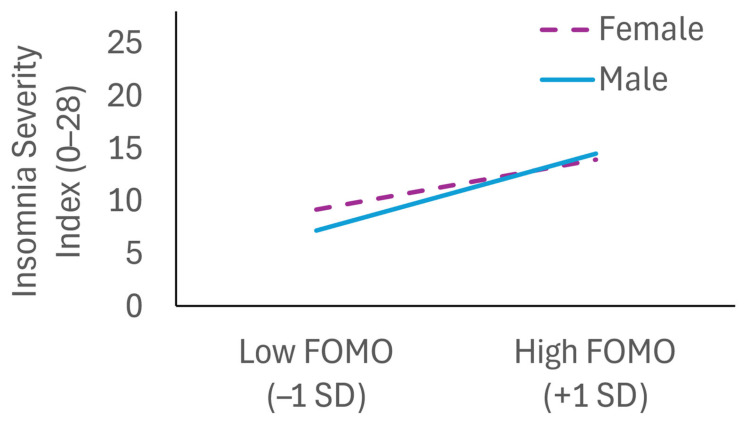
Interaction of gender and fear of missing out (FOMO) predicting insomnia symptoms (Insomnia Severity Index score). Covariate-adjusted simple slopes are positive and statistically significant for both female and male participants, with a steeper slope among males.

**Table 1 brainsci-15-00917-t001:** Demographic characteristics and key study variables stratified by gender. Data are presented as means and standard deviations or percentages.

Characteristic	Female (*n* = 535)	Male (*n* = 304)	Non-Binary (*n* = 10)
Age	23.9 (3.9)	24.8 (3.8)	24.4 (3.7)
Race/Ethnicity			
White	56.8	47.0	70.0
Black or African American	25.2	34.5	30.0
Hispanic or Latino	16.6	13.8	20.0
Asian or Asian American	4.7	5.6	0.0
American Indian or Alaska Native	2.1	3.6	30.0
Native Hawaiian or Other Pacific Islander	0.4	0.0	10.0
Another Race/Ethnicity	0.7	0.7	0.0
Education	2.9 (1.2)	3.0 (1.3)	2.6 (1.4)
Fear of Missing Out	2.6 (0.9)	2.7 (0.9)	2.8 (1.1)
Insomnia Severity	11.4 (6.0)	11.1 (6.3)	13.9 (4.7)
Depressive Symptoms	13.3 (6.0)	12.9 (6.1)	15.4 (7.2)

## Data Availability

The data for this study are available at https://osf.io/af6ve/ (accessed on 3 August 2025).

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
