# Peer review of "Insomnia as a Behavioral Pathway from Fear of Missing Out to Depression in Emerging Adults"

_brainsci, 2025, doi:10.3390/brainsci15090917_

Round 1

Reviewer 1 Report

Comments and Suggestions for Authors

Overall, this was a very well written paper. However, I have a couple of comments, largely relating to the theoretical framework and clinical implications suggested by the authors. 

  1. Although the theoretical frameworks discussed on the introduction are interesting, is there any empirical evidence supporting these claims as the basis of FOMO? As it stands there is a lot of discussion of different theories that are not then tested by this study. The introduction should be streamlined to focus on the salient aspects that are being addressed by this study.
  2. The authors state in the discussion that ‘ because we observed that insomnia partial mediated the link between FOMO and depressive symptoms, improving sleep may be a particularly viable and promising intervention target…’. However, I am unclear what the proposed benefit is thought to be by improving sleep based on this study. Is it that improving sleep is thought to reduce FOMO or is it to reduce the possible consequences of FOMO? Either way I think consideration is needed as to the potential implications of the findings of this study.
  3. In the limitation section – the self-reported measure is for insomnia symptoms not sleep disruption per se, so objective assessment of sleep would not really provide a better measure of insomnia since insomnia is a subjective disorder, instead clinical diagnosis would be a better option.
  4. Finally, where there any other inclusion criteria for the original study where the data was collection other than age? Just to help understand the population included in this study.

Author Response

Reviewer 1

  1. Although the theoretical frameworks discussed on the introduction are interesting, is there any empirical evidence supporting these claims as the basis of FOMO? As it stands there is a lot of discussion of different theories that are not then tested by this study. The introduction should be streamlined to focus on the salient aspects that are being addressed by this study.

We appreciate this suggestion. Because Reviewer 2 highlighted the theoretical framing as a strength, we retained the introduction’s structure but streamlined the scope by adding a sentence on Lines 65–68 clarifying that these frameworks function as conceptual scaffolds for a single, testable mechanistic pathway examined in this study (FOMO → insomnia → depressive symptoms; gender moderation).

  1. The authors state in the discussion that ‘because we observed that insomnia partial mediated the link between FOMO and depressive symptoms, improving sleep may be a particularly viable and promising intervention target…’. However, I am unclear what the proposed benefit is thought to be by improving sleep based on this study. Is it that improving sleep is thought to reduce FOMO or is it to reduce the possible consequences of FOMO? Either way I think consideration is needed as to the potential implications of the findings of this study.

Thank you for this helpful point. We revised the discussion to clarify the potential implications. Please see Lines 313–318.

  1. In the limitation section – the self-reported measure is for insomnia symptoms not sleep disruption per se, so objective assessment of sleep would not really provide a better measure of insomnia since insomnia is a subjective disorder, instead clinical diagnosis would be a better option.

We agree with the reviewer, and we revised the limitations to clarify that a clinician-rated diagnosis would be the appropriate criterion assessment for insomnia. Please see Lines 341–345.

  1. Finally, where there any other inclusion criteria for the original study where the data was collection other than age? Just to help understand the population included in this study.

We clarified the eligibility criteria on Lines 134–137 and confirmed that no additional inclusion or exclusion criteria were applied beyond age, United States residency, and English fluency.

Reviewer 2 Report

Comments and Suggestions for Authors

The study examined the role of insomnia as a mediating factor between fear of missing out (FOMO) and depression symptoms among emerging adults in the US and also examined whether gender moderated the relations between FOMO and insomnia and between FOMO and depression. Results of this cross-sectional study conducted with adults 18-30 years old found that insomnia partially mediated the relationship between FOMO and depression, and that the relationship between FOMO and insomnia symptoms was greater for participants identifying as male vs female. This was an interesting study and I applaud the use of mediation and moderation analyses to better understand the interrelationships between these variables. I offer the suggestions and recommendations below to further strengthen and add value to this paper. 

Abstract

I suggest describing the term “emerging adults” at first mention as not all readers will be familiar with this term.

Introduction

The introduction is well-written and grounded in theoretical models to support the study aims and hypotheses.

Methods

  • 1. Measures – please mention the directionality of each measure. Are higher scores indicative of higher or lower FOMO, insomnia severity, and depression symptoms?
  • 2. Data analysis - As not everyone is familiar with mediation and moderation analyses, I suggest describing these analyses a little more
  • Why not test for a moderated mediation model? Rather than running the mediation and moderation analyses separately? I understand those identifying as non-binary were removed from the moderation analysis, but at a minimum, it would be interesting to see a sensitivity analysis where the mediation model is run separately for males and females.

Results

  • Were other demographic or health characteristics collected? It would be good to see information on other characteristics, such as marital status, occupation, and diagnosed health conditions
  • I suggest including a table showing demographic characteristics stratified by gender.
  • Please report the FOMO, insomnia severity, and depression symptom scores stratified by gender. I think this is important to include since you examined gender as a moderator – how did these scores differ by gender?
  • I highly suggest adding a figure to show the interaction of gender and FOMO in predicting insomnia – it would add value and aid in interpreting the moderation analysis results. From what I can see, the relationship between FOMO and insomnia is in the same direction regardless of gender identify; it’s just that the association is stronger for males.
  • As stated above in the methods, I would be curious to see the results if this was run as a moderated mediation model.

Discussion

  • Line 263-264 mentions that men may be more likely to respond to perceived exclusion with withdrawal or rumination. However, to my knowledge, women are more likely to ruminate not men, so suggest rethinking this statement. There’s also research showing that co-rumination has more negative impact on females than males.
  • Also, the argument presented in this paragraph (258-267) lines suggests that there is a positive association between FOMO and insomnia symptoms for males but not for females. But the results reported shows that there was a positive, statistically significant association between FOMO and insomnia for both males and females. It would be good to make sure this is clear in the discussion. It may be that those who identify as female do manage social stressors through affiliative behaviors, which can lessen the impact of FOMO on insomnia, but they may also engage in behaviors (e.g., rumination - brooding, excessive reassurance thinking) that may exacerbate insomnia symptoms associated with FOMO (albeit to a lesser extent than males).
  • Suggest removing the clause “but not depression” (line 258) from this sentence as your aren’t addressing why the association between FOMO and insomnia, but not depression, is moderated by gender: “Finally, the gender moderation observed for insomnia, but not depression, suggests that individual differences in coping responses to perceived social threat may shape how FOMO impacts health and well-being.”
  • Do you have any thoughts or speculation on why we don't see that gender moderates the association between FOMO and depression?
  • In the limitations section, the authors address other mechanisms than insomnia that could drive relation between FOMO and depression – but prior to that, I do not see much discussion about the fact that insomnia was only a partial mediator, there’s still a direct effect of FOMO on depressive symptoms not explained by insomnia. Are the authors suggesting these as other potential mediators between FOMO and insomnia?

Author Response

Reviewer 2

  1. Abstract: I suggest describing the term “emerging adults” at first mention as not all readers will be familiar with this term.

Thank you. We revised the abstract to define emerging adults at first mention.

  1. Introduction: The introduction is well-written and grounded in theoretical models to support the study aims and hypotheses.

We appreciate the positive comment and retained the introduction’s theoretical framing.

  1. Measures: Please mention the directionality of each measure. Are higher scores indicative of higher or lower FOMO, insomnia severity, and depression symptoms?

Thank you. We clarified the directionality of all measures in Section 2.1: higher scores indicated greater FOMO, insomnia severity, and depressive symptoms, respectively.

  1. Data analysis - As not everyone is familiar with mediation and moderation analyses, I suggest describing these analyses a little more.

We expanded Section 2.2 with a brief, plain-language description of the mediation, moderation, and moderated-mediation analyses.

  1. Why not test for a moderated mediation model? Rather than running the mediation and moderation analyses separately? I understand those identifying as non-binary were removed from the moderation analysis, but at a minimum, it would be interesting to see a sensitivity analysis where the mediation model is run separately for males and females.

Thank you for this suggestion. We added an exploratory moderated-mediation analysis to the manuscript as Section 3.3.

We found that the conditional effect of FOMO on depressive symptoms via insomnia symptoms was significant for both females and males, but larger among males with a significant index of moderated mediation. These findings converged with the primary mediation and moderation results.

  1. Were other demographic or health characteristics collected? It would be good to see information on other characteristics, such as marital status, occupation, and diagnosed health conditions

Thank you for raising this. These variables were not collected in the parent study. We have clarified this in the limitations section. Please see Lines 354–356.

  1. I suggest including a table showing demographic characteristics stratified by gender. Please report the FOMO, insomnia severity, and depression symptom scores stratified by gender. I think this is important to include since you examined gender as a moderator – how did these scores differ by gender?

We added Table 1 presenting demographic characteristics and key study variables stratified by gender, including FOMO, insomnia severity, and depressive symptoms.

  1. I highly suggest adding a figure to show the interaction of gender and FOMO in predicting insomnia – it would add value and aid in interpreting the moderation analysis results. From what I can see, the relationship between FOMO and insomnia is in the same direction regardless of gender identify; it’s just that the association is stronger for males. As stated above in the methods, I would be curious to see the results if this was run as a moderated mediation model.

We added Figure 2, a simple slopes plot depicting the association between FOMO and insomnia symptoms separately for female and male participants, from models adjusted for age, race, and education. As noted by the reviewer, both slopes are positive with a steeper slope among males.

  1. Discussion Line 263-264 mentions that men may be more likely to respond to perceived exclusion with withdrawal or rumination. However, to my knowledge, women are more likely to ruminate not men, so suggest rethinking this statement. There’s also research showing that co-rumination has more negative impact on females than males.

Thank you for this helpful correction. We revised the discussion to avoid implying that men ruminate more than women. Please see Lines 289–307.

  1. Also, the argument presented in this paragraph (258-267) lines suggests that there is a positive association between FOMO and insomnia symptoms for males but not for females. But the results reported shows that there was a positive, statistically significant association between FOMO and insomnia for both males and females. It would be good to make sure this is clear in the discussion. It may be that those who identify as female do manage social stressors through affiliative behaviors, which can lessen the impact of FOMO on insomnia, but they may also engage in behaviors (e.g., rumination - brooding, excessive reassurance thinking) that may exacerbate insomnia symptoms associated with FOMO (albeit to a lesser extent than males).

We have corrected this paragraph to reflect that FOMO was positively associated with insomnia symptoms for both males and females, with a stronger association among males.

  1. Suggest removing the clause “but not depression” (line 258) from this sentence as your aren’t addressing why the association between FOMO and insomnia, but not depression, is moderated by gender: “Finally, the gender moderation observed for insomnia, but not depression, suggests that individual differences in coping responses to perceived social threat may shape how FOMO impacts health and well-being.”

We made this change.

  1. Do you have any thoughts or speculation on why we don't see that gender moderates the association between FOMO and depression?

Thank you for this suggestion. We added several sentences to the discussion on Lines 300–307 speculating on why gender did not moderate the link between FOMO and depression.

  1. In the limitations section, the authors address other mechanisms than insomnia that could drive relation between FOMO and depression – but prior to that, I do not see much discussion about the fact that insomnia was only a partial mediator, there’s still a direct effect of FOMO on depressive symptoms not explained by insomnia. Are the authors suggesting these as other potential mediators between FOMO and insomnia?

We now state explicitly that insomnia partially mediated the FOMO and depressive symptoms association, leaving a residual direct effect, and suggest other potential mediators that warrant testing in future research. Please see Lines 358–361.
